**Subject Category:**
Biology (whole organism)

ecology/evolution

sexual selection, reproductive allocation, potential reproductive rate, nest-site availability, food abundance, Gobiidae

**Author for correspondence:**
Aurora García-Berro
e-mail: auroragbn@gmail.com

# Understanding resource driven female–female competition: ovary and liver size in sand gobies

Aurora García-Berro[1,2], Johanna Yliportimo[3],
Kai Lindström[3] and Charlotta Kvarnemo[2,4]

[1]Department of Evolutionary Biology, Ecology and Environmental Sciences, University of Barcelona, Barcelona, Spain
[2]Department of Biological and Environmental Sciences, University of Gothenberg, Gothenberg, Sweden
[3]Environmental and Marine Biology, Åbo Akademi University, Turku, Finland
[4]Linnaeus Centre for Marine Evolutionary Biology, University of Gothenburg, Gothenburg, Sweden

AG-B, 0000-0002-2419-2516; KL, 0000-0002-8356-5538;
CK, 0000-0001-8983-2900

The operational sex ratio (OSR, ready-to-mate males to females) is a key factor determining mating competition. A shortage of a resource essential for reproduction of one sex can affect OSR and lead to competition within the opposite sex for resource-holding mates. In the sand goby (*Pomatoschistus minutus*), a fish with paternal care, male readiness to mate depends on acquiring a nest-site, whereas food abundance primarily impacts female egg production. Comparing body condition and gonadal investment of fish from two populations with different availability in resources (Baltic Sea: few nest-sites, more food; North Sea: many nest-sites, less food), we predicted females carrying more mature eggs in the Baltic Sea than in the North Sea. As predicted, ovaries were larger in Baltic Sea females, and so was the liver (storage of energy reserves and vitellogenic compounds) for both sexes, but particularly for females. More females were judged (based on roundness scores) to be ready to spawn in the Baltic Sea. Together with a nest colonization experiment confirming a previously documented difference between the two areas in nest-site availability, these results indicate a more female-biased OSR in the Baltic Sea population, compared to the North Sea, and generates a prediction that female–female competition for mating opportunities is stronger in the Baltic population. To our knowledge, this is the first time that female reproductive investment is discussed in relation to OSR using field data.

# 1. Introduction

Resource availability is known to differentially affect the sexes' readiness to mate, thus biasing the operational sex ratio (OSR; [1]). In particular, when a resource is scarce, fewer individuals from the resource-limited sex become 'qualified to mate' (sensu [2]), and the potential for mate choice increases for that sex, while competition for resource-holding mates is predicted to increase within the opposite sex. This can create uncertainty for future reproduction, which may select for higher current reproductive investment [3], as predicted by life-history theory [4]. However, while many studies have investigated effects of limited mate availability on behavioural traits (e.g. [5]), surprisingly little attention has been given to life-history related reproductive traits.

In many fish species with paternal care, a nest-site is an essential resource for male reproduction, as it provides a place for egg deposition (e.g. [6]). Consequently, in environments with low nest-site abundance there will be a shortage of ready-to-mate males (nest-holders with space for eggs), leading to a female biased OSR [2,7,8]. Similarly, because reproducing females in many taxa have a higher dependence on food intake than reproducing males (e.g. [9,10]), food availability often limits egg maturation rate, affecting female potential reproductive rate and OSR [11]. In such systems, high food abundance and low nest-site availability are predicted to contribute to a female-biased OSR, which represents a diminished chance for females to gain future access to mates.

The sand goby, *Pomatoschistus minutus*, is a fish with one breeding season, during which it breeds repeatedly in multiple brood cycles [9]. Females are batch-spawners, meaning they lay all their mature eggs at once, then spend one to two weeks unable to spawn until a new batch of eggs has matured [12]. Whether a female is ready to spawn or not can be estimated visually based on her roundness [13]. Males show paternal care by building nests under mussel shells. Once the nest is full he fans and guards the eggs until they hatch one to three weeks later [12,14].

Nest-site availability has been shown to fundamentally affect OSR and sexual selection in sand gobies [15], and female potential reproductive rate to be strongly affected by food availability [9]. Here, we return to the same two populations studied in [15], and compare fish from these geographically distant areas that differ markedly in nest-site abundance, and arguably also in food availability (see Material and methods). We predict to find more ready-to-mate females (and hence a more female-biased OSR), that both sexes will show higher condition, and that females will allocate more resources to reproduction (ovaries) in the population characterized by high prey and low nest-site abundance, compared to individuals from the low prey and high nest-site population.

# 2. Material and methods

The study sites are the Swedish west coast close to the North Sea (Bökevik; 58°14′53.2″ N 11°26′49.7″ E; 20–30‰ salinity) and the Finnish south coast of the Baltic Sea (Tvärminne; 59°49′25.3″ N 23°08′35.1″ E; 6–7‰ salinity). We refer to these locations as 'North Sea' and 'Baltic Sea', respectively, from here on. Previous work has shown that nest-site availability (empty mussel shells) differs markedly between these areas [15,16], with 48 times as many nest-sites per m$^2$ at the North Sea location compared to the Baltic Sea location (mean ± s.e.: 9.6 ± 2.3 and 0.2 ± 0.03 nest-sites per m$^2$ respectively) [15]. In contrast, food availability is likely to be lower at the North Sea than the Baltic Sea sites. Sand gobies eat primarily benthic macrofauna, including infauna and small free-swimming crustaceans, such as mysids [17,18]. The eutrophication status in the coastal waters of southwest Finland is generally higher compared to the Swedish west coast [19], affecting the benthic invertebrate abundance in the two sites, reviewed in electronic supplementary material, table S1 [17,20–25]. In addition, based on our own observations, mysids are often found in exceptionally high densities in the shallow areas where sand gobies breed at the Baltic Sea site, but not at the North Sea site. We therefore predict these differences in nest-site availability and food availability to affect the OSR.

Using a hand trawl, sand gobies were caught at the peak of their breeding season (early June 2014 in the North Sea location, late June 2014 in the Baltic Sea location, thus adjusting for a later start and peak of the breeding season at the latter site). Eighteen females and 23 males from the North Sea site, and 31 females and 28 males from the Baltic site were caught. No selective criteria were applied, i.e. all fish caught were kept. The fish were euthanized (2-phenoxyethanol 1 ml l$^{-1}$), measured for total length (TL) and preserved in 95% ethanol until dissections. North Sea females were 54.3 ± 1.2 mm (TL, mean ± s.e.) and males 52.3 ± 1.3 mm; Baltic Sea females were 50.9 ± 1.6 mm and males 48.8 ± 1.4 mm.

All dissected organs were dried at 70°C for greater than or equal to 24 h before being weighed to the nearest 0.01 mg.

Whether a female is ready to spawn or not can be estimated visually based on her roundness [13]. The fish were photographed directly after they were euthanized, and based on the photographs, each female was scored according to a 'reproductive maturity index' (RMI), ranging from 0 to 3 (electronic supplementary material, figure S1). A female given an RMI value of 2.25 or higher is judged as being ready to spawn.

To investigate whether the estimates of nest-site availability in [15], collected 26 years ago, are still relevant, we repeated the same nest-site colonization experiment (8–11 June 2014 in the North Sea location, 1–4 July 2019 in the Baltic Sea location). We placed 20 artificial nest-sites in shallow water (50–60 cm depth) and checked colonization by males after 24, 48 and 72 h. Nest-sites consisted of halved clay flower pots of 65 mm diameter, which were placed 2 m apart, in two rows of 10 pots and separated by a distance of greater than or equal to 5 m. Nest building activity was rated each time from 0 (no signs of building) to 5 (completely covered nest). A nest-site was considered colonized if the nest was scored 2–5. Presence of eggs was determined at the end of the experiment, that is after the 72 h check.

Rate of nest-site colonization was compared using a Kolmogorov–Smirnov test. The proportion of nests with eggs in them, and the proportion of ready-to-spawn females were compared using Fisher's exact test. RMI-values of the two populations were based on scored values, and therefore analysed using a Mann–Whitney $U$-test. All other data were checked for homoscedasticity and normality, transformed if deviating, and analysed using parametric tests. All non-significant interactions were removed from the models. Dry body mass (somatic, with liver, ovaries and intestines removed) was used to estimate body size. Condition was measured as dry liver mass with dry body mass as covariate to control for body size and analysed using analysis of covariance (ANCOVA; factors: population and sex; covariate: body mass). ANCOVA was used to avoid potential problems associated with indexes and residuals [26]. Nevertheless, to allow comparison with other studies we also calculated the hepatosomatic index (HSI = dry liver mass × 100/dry body mass), and tested it using two-way analysis of variance (ANOVA; factors: population and sex). Similarly, if female gonadal investment was measured as dry ovary mass, with dry body mass as covariate to control for body size and analysed using ANCOVA (factor: population; covariate: body mass). We also calculated the gonadosomatic index (GSI = dry ovary mass × 100/dry body mass) to allow comparison with other studies. One ovary (North Sea) was lost during handling, reducing the sample size for ovary mass and GSI. Body, liver, ovary dry mass and GSI were also compared using one-way ANOVAs (factor: population). Analyses were performed using SPSS 22.0.

## 3. Results

We found a faster nest-site colonization rate in the Baltic Sea than in the North Sea location (Kolmogorov–Smirnov test statistic = 1.732, $p = 0.005$). After 72 h, all artificial nest-sites were colonized in the Baltic Sea location, but only 10 of 20 nest-sites were colonized in the North Sea location (figure 1). All but one of the nests contained eggs in the Baltic Sea location, whereas no eggs were found in any of the nests in the North Sea (Fisher's exact test: $p < 0.001$).

Based on our RMI-values, female roundness differed between the two populations, with rounder females in the Baltic Sea (mean ± s.e. RMI-value [mean rank]: North Sea: 1.52 ± 0.13 [19.2]; Baltic Sea: 2.32 ± 0.10 [35.0]; Mann–Whitney $U$-test; $U = 99.0$, $p < 0.001$). The proportion of females that were scored as being ready to spawn (i.e. had a RMI-value ≥ 2.25) was also higher in the Baltic Sea (North Sea: 6 of 31 = 0.19; Baltic Sea: 13 of 18 = 0.72; Fisher's exact test: $p < 0.001$).

In our sample, neither females nor males differed between populations in somatic dry body mass (table 1). For both sexes, but particularly for females, dry liver mass was higher in fish from the Baltic Sea compared to the North Sea, both when analysed with dry body mass as covariate (square-root transformed: ANCOVA; population: $F_{1,95} = 90.78$, $p < 0.001$, sex: $F_{1,95} = 59.17$, $p < 0.001$: body mass: $F_{1,95} = 78.52$, $p < 0.001$; population × sex: $F_{1,95} = 13.13$, $p < 0.001$) and as HSI (square-root transformed; two-way ANOVA; population: $F_{1,96} = 104.4$, $p < 0.001$; sex: $F_{1,96} = 67.66$, $p < 0.001$; population × sex: $F_{1,96} = 19.18$, $p < 0.001$; figure 2). In absolute measures, however, only female dry liver mass differed significantly between the two populations, whereas male dry liver mass did not (table 1).

Similarly to liver mass, dry ovary mass was positively influenced by dry body mass and, consistent with the RMI-values, it was significantly higher for Baltic than North Sea females (square-root

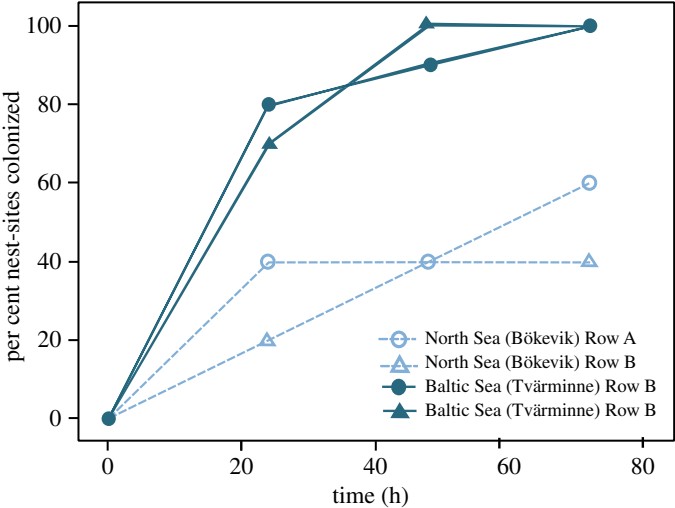

**Figure 1.** Colonization by male sand gobies, *Pomatoschistus minutus*, of 20 artificial nest-sites placed in two rows at each of our study locations. The nest-sites were inspected after 24, 48 and 72 h, as marked by open symbols and dashed lines for the North Sea location, and filled symbols and solid lines for the Baltic Sea location.

**Table 1.** Measured traits (mean ± s.e.) of female and male sand gobies, *Pomatoschistus minutus*, from a North Sea and a Baltic Sea population.

|  | North Sea (Bökevik) | Baltic Sea (Tvärminne) | F | p-value |
|---|---|---|---|---|
|  |  |  | $F_{1,47}$ | p-value |
| [a]female somatic body mass (mg) | 155.80 ± 11.53 | 130.07 ± 11.85 | 2.13 | 0.15 |
| [b]female liver mass (mg) | 2.69 (2.33; 3.09) | 7.50 (6.59; 8.47) | 30.0 | <0.001 |
|  |  |  | $F_{1,49}$ | p-value |
| [a]male somatic body mass (mg) | 159.99 ± 11.14 | 138.81 ± 12.29 | 1.63 | 0.21 |
| [c]male liver mass (mg) | 1.44 (1.25; 1.65) | 2.34 (1.89; 2.89) | 3.88 | 0.055 |
|  |  |  | $F_{1,46}$ | p-value |
| [a]ovary mass (mg) | 12.90 ± 2.88 | 25.11 ± 4.21 | 6.10 | 0.017 |
| [b]female gonadosomatic index (GSI) | 5.53 (4.43; 6.98) | 18.36 (15.46; 21.51) | 17.4 | <0.001 |

All measures are based on dry mass. Analyses were done using [a]untransformed, [b]square-root transformed or [c]Ln-transformed data. For [a] untransformed mean ± s.e. values are given. For [b] and [c] backtransformed values are given, reported as mean (lower range of s.e.; higher range of s.e.).

transformed; ANCOVA; population: $F_{1,45} = 14.04$, $p = 0.001$; body mass: $F_{1,45} = 6.91$, $p = 0.012$; figure 3). GSI was also significantly higher for Baltic than North Sea females (table 1), and minimum dry ovary mass followed the same pattern (figure 3).

## 4. Discussion

Focusing on sand gobies from two populations that differ markedly in food and nest-site availability [15,16,19,23,24], as predicted, we found a greater proportion of females being ready-to-spawn in the Baltic Sea population, and female ovary and liver size (both sexes) to be larger in fish from the Baltic Sea (high food, low nest availability), compared to the North Sea (opposite pattern). Our data on nest-site colonization and spawning rates (figure 1), together with a previously documented difference between the two areas in nest-site availability [15], suggest a more female-biased OSR in the Baltic Sea population, compared to the North Sea. As such, it also generates a prediction that female–female competition for mating opportunities is stronger in the Baltic population [5,15,27].

The high condition found in Baltic Sea fish (figure 2) is consistent with the historical eutrophication of the Kattegat area and the Baltic Sea proper investigated in Andersen *et al.* 2017 [19] and references

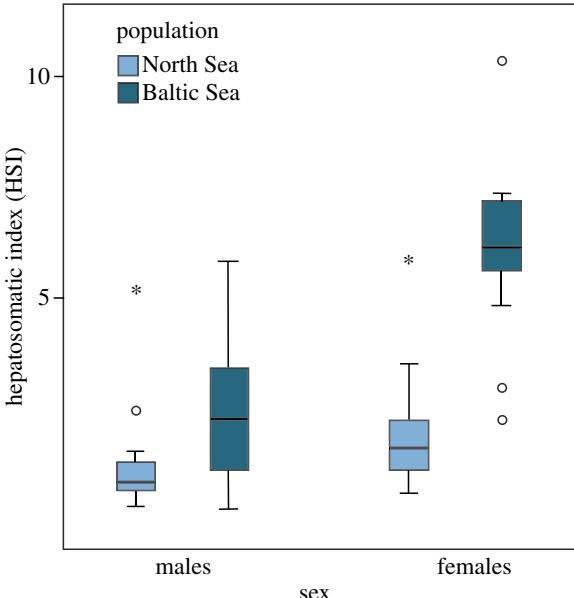

**Figure 2.** Hepatosomatic index (HSI: dry liver mass × 100/dry gutted somatic body mass) of male and female sand gobies, *Pomatoschistus minutus*, from a North Sea and a Baltic Sea population. Untransformed values are shown here, while square-root transformed data were used in the analysis. In the graph, bands indicate median values, boxes represent 25th to 75th percentiles and whiskers represent 95th percentiles. Outliers and extreme outliers are shown as circles and asterisks respectively.

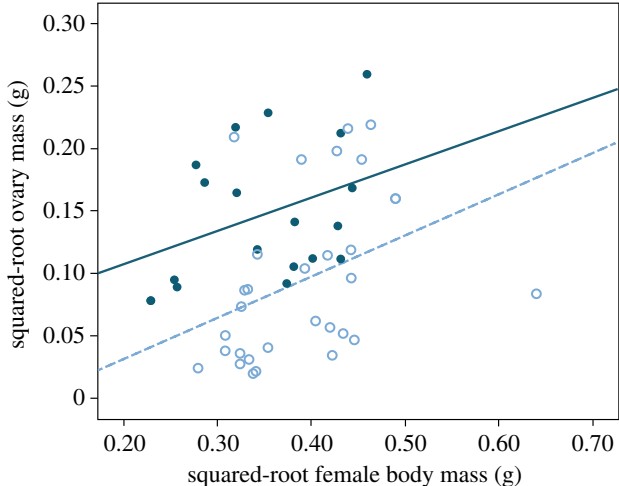

**Figure 3.** Ovary mass in relation to body mass (somatic gutted body mass) in female sand gobies, *Pomatoschistus minutus*, from a North Sea population (open circles: sqrt(dry ovary mass) = [−0.03 + 0.33 × sqrt(dry body mass)]) and a Baltic Sea population (closed circles: sqrt(dry ovary mass) = [0.06 + 0.26 × sqrt(dry body mass)]).

therein, and with our own observation that especially mysids are found in very high densities in the bays where sand gobies breed in this area. However, liver has a dual function in females, and the significant interaction between sex and location probably reflects the fact that the liver does not only store energy reserves, but also synthesizes vitellogenin (a protein, the plasma precursor of yolk) [28]. Gonadal development and vitellogenesis affect the liver along the reproductive cycle and is reflected in both HSI and ultrastructure of the hepatocytes [29,30]. Thus, the observed high HSI may indicate both a better condition and more vitellogenin synthesized in Baltic Sea than in North Sea fish.

Salinity affects the need for osmoregulation, and hence could be a source of physiological stress in this species, potentially contributing to the found difference in relative liver size between sand gobies from the two localities. Fish are known to compensate for osmoregulatory costs by allocating less energy to growth, with highest growth rates observed in salinity close to isotonic levels (9‰) [31]. However, due to its marine ancestry, sand goby condition may be expected to be lower in the 6–7‰ found at our Baltic Sea site, compared to the 20–30‰ typical for our North Sea site. Consistent with this

expectation, the close relative common goby (*Pomatoschitus microps*) shows a decrease in body size with decreasing salinity along the Baltic basin [16]. Interestingly, however, body size did not differ in our samples and, importantly, liver size followed the opposite pattern, and with marked sex-differences as shown in figure 2. Moreover, Baltic and North Sea sand gobies that were kept in 6‰ or 30–32‰ salinity for 2.5–4 months did not differ in condition and growth [32]. Therefore, it remains unclear whether osmoregulation is a source of physiological stress in this species, and to what extent (if at all) it may have contributed to the larger relative liver size in Baltic Sea fish.

We found that Baltic Sea females had larger ovaries than North Sea females, both in absolute and relative terms. It is likely that larger ovary size translates into larger clutch size, but whether it also affects egg size or the time between successive clutches laid in this species are interesting questions that could be addressed in future studies. The larger ovaries from the Baltic Sea fish are likely a result of higher food availability (cf. above), combined with limited spawning opportunities in the Baltic Sea region [15]. Females experiencing a female-biased OSR may therefore build up a surplus of mature eggs in their ovaries. Interestingly, common goby females have been shown to lay larger clutches under an experimentally female-biased OSR [3]. Heubel *et al.* [3] predicted that if the chance of finding a nest-holding male in the future is low, the best solution for females is to maximize current reproductive success by laying many eggs at once, a situation which is likely to select for large ovary size. Our results may therefore reflect a phenotypic adjustment in female life-history, or even a genetic adaptation, with allocation to fecundity depending on food and mate availability. Another non-exclusive explanation is that female sand gobies with large ovaries are preferred by males [33]. Since the Baltic Sea population arguably reproduces under a more female-biased OSR, males can potentially be choosier and females might have to compete for mating opportunities by being more attractive. Moreover, Baltic Sea females had much larger minimum ovary sizes than females from the North Sea (figure 3). These values are very likely to represent females caught shortly after spawning, hence, being a measure of mature but empty ovary sizes. Therefore, as a speculation, we suggest that ovary size may have evolved in response to sexual selection for larger ovaries in the Baltic Sea.

These findings should be interpreted on a local basis, consistent with the limitation of two sampling sites, although it would be interesting to test if these effects of food availability and spawning opportunities on female gonad size are a general trend in this species. In addition, other selective agents may affect ovary size among animal populations. Here, we take a closer look at three such agents. (i) For short-lived species, like sand gobies, a short breeding season may select for larger ovary size, as a way to produce as many eggs as possible in a short time. However, sand gobies are known to breed in 6–19°C [34] and despite a later start in the Baltic Sea they also stop breeding later compared to the North Sea, which results in a roughly two-month breeding season in both areas. Thus, length of breeding season is unlikely to explain the found difference in ovary size. (ii) In flounder, low salinity populations produce eggs with larger diameter and lower gravity [35]. However, egg dry mass does not differ between areas that differ in salinity, showing that the larger diameter in low salinity is generated by higher water content [35]. Thus, even if sand goby females may also have larger egg diameter in the Baltic Sea, it would be unlikely to explain our finding of higher ovary dry mass in Baltic Sea females. (iii) Compared to the Baltic Sea, predation risk on adults may be higher in the North Sea, potentially creating a stronger selection against becoming overly round before spawning a clutch there. Yet, adult females of both common and two-spotted goby in the same North Sea area are often extremely round, making a difference in predation pressure a questionable explanation for the found differences in ovary size, as these species are presumably exposed to similar predation pressure. Both common and two-spotted goby females in this area are known to have limited spawning opportunities [8,36], which further support our notion that this variable affects relative ovary size. Nevertheless, both field and experimental data on length of breeding season, diameter and dry mass of sand goby eggs spawned in different salinities, and predation risk in relation to female roundness should be collected in future studies, before they can be fully ruled out.

In conclusion, our field data are consistent with predictions from OSR theory. Large differences in female reproductive investment were found in these two environments, presumably as a result of differences in mating competition, although other selective pressures that were not considered here might also be in effect. Further behavioural observations are needed to better understand the underlying reasons, along with a direct quantification of OSR for the two areas.

Ethics. Our work was done in accordance with national laws in Sweden and Finland. It was approved by the Ethical Committee for Animal Research in Gothenburg Sweden (permit Dnr 86-2013) and Finland (following regulations by the Animal Experiment Board of Finland).

Data accessibility. Data are available from the Dryad Digital Repository at: https://doi.org/10.5061/dryad.51j892v [37].
Authors' contributions. C.K. conceived study, collected fish and 2014 colonization data; A.G.-B. dissected fish and collated data; J.Y. collected 2019 colonization data; A.G.-B., C.K. and K.L. analysed the data; A.G.-B. and C.K. wrote the paper, with input from J.Y. and K.L. All authors approved the final manuscript version and agree to be held accountable for its content.
Competing interests. We declare we have no competing interests.
Funding. Swedish Research Council (grant no. 621-2011-4004) to C.K.
Acknowledgements. We thank Ingrid Ahnesjö for valuable comments, Alf Norkko for additional references on prey abundance and Sami Merilaita for help generating figures.

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
