## [Reviewer comments · Royal Society Open Science]

Review History

RSOS-181460.R0 (Original submission)

Review form: Reviewer 1

Is the manuscript scientifically sound in its present form?

No

Are the interpretations and conclusions justified by the results?

Yes

Is the language acceptable?

Yes

Is it clear how to access all supporting data?

Yes

Do you have any ethical concerns with this paper?

No

Have you any concerns about statistical analyses in this paper?

No

Recommendation?

Reject

Comments to the Author(s)

This paper uses a well-established study system (sand goby fish) with a contrast in nesting sites between two regions of its range: 1. high density of nesting sites in the North Sea) and 2. their shortage in the Baltic Sea. This contrast has been used for many previous studies on sexual selection, providing strong background information to which the current study can relate. Baltic and North Seas certainly differ in many other aspects and one of them is contrasting food availability. Based on that, the authors predict (or expect as they phrase it) more female-female competition for reproductive opportunities in the well fed but nest site limited females in the Baltic. Indeed, they found heavier ovaries and liver in Baltic sand goby females, with similar, though weaker trend in the liver of males (testes were not measured).

It is a straightforward paper, with adequate sample size. The finding is not surprising (well-fed females have larger ovaries and liver – a fat storage organ in many fishes) and it provides the main strength and weakness of the study.

Strength: outcome is clear-cut; contrast between two populations provide difference that is then discussed.

Weakness: comparing only two contrasting populations. It may be read as a preliminary insight into a broader interesting study and the authors do not help to avoid this feeling. For example, in discussion it is clear that they struggle to discount alternative explanations. In many cases they succeed (the last paragraph) but most of discussion is composed in taking one potential explanation after another and discussing its potential to explain that contrast between the populations. A more useful approach would be to perform a follow-up experimental work to test competing hypotheses arising from this study.

Comments:

Introduction is succinct and well written.

Methods would certainly deserve more details on the contrast in prey abundance, density etc. This is critical for the paper but the general reference to an inventory of benthic communities should be clarified in what aspect the two sites differ (food abundance, availability, size, diversity, presence (or abundance) of one particularly important prey taxon etc.). It is addressed in part in discussion (line 122) but in a very vague way – “our own observations ... mysids extremely abundant...”

The same applies to nest site availability – is it 20% difference? Or 10 fold difference? I can imagine it is quantified in the papers that authors refer to, but reader needs to get this simple information without downloading other papers.

Results are fine, though more information is needed (perhaps in methods) on why “absolute measures” were tested in a different way and how the two measures used are useful.

Discussion rests on explaining potential alternative explanations. In some parts, it reads overly defensive. As a reader I would feel that referees made authors to write all those long paragraphs. It is of course very good to list all alternative factors that may be behind observed contrasts, but

one or two sentences for discounting them (if you have strong arguments against, and in many cases you do) would suffice.

An example of this overly defensive style: Line 118, ...suggesting that female-female competition for mating opportunities may be stronger in the Baltic... - the three cited references arrived to the same conclusion, so I think it is a further confirmation rather than “suggesting that”

One particular case that made me wonder why this has not been part of the study (among others) is paragraph starting on line 145. It is on egg size in female with large/small ovaries. When ovary mass was measured, it was very easy to measure a subset of eggs per female. Why at least this has not been done?

Table 1. Why morphological traits? Perhaps life history traits? I think morphological denotes body dimensions etc.

Also, mean values are given for untransformed data – why not back-transformed data when ln- and sqrt-transformation was used in the tests?

Review form: Reviewer 2

Is the manuscript scientifically sound in its present form?

Yes

Are the interpretations and conclusions justified by the results?

Yes

Is the language acceptable?

Yes

Is it clear how to access all supporting data?

Yes

Do you have any ethical concerns with this paper?

No

Have you any concerns about statistical analyses in this paper?

I do not feel qualified to assess the statistics

Recommendation?

Major revision is needed (please make suggestions in comments)

Comments to the Author(s)

In their manuscript ‘Understanding resource driven female-female competition: ovary and liver size in sand gobies’ (RSOS-181460) the authors report on their investigation of physiological differences between two populations of sand gobies. Because these populations differ in ecological traits related to reproduction, the authors focus on population-specific characteristics of female reproductive investment. Crucially, they state that one of the populations features traits that should make reproduction relatively more easy for males (many nest sites), while the other should make reproduction relatively more easy for females (high food abundance). They predict that these differences should (i) alter the operational sex ratio, (ii) these differences in the OSR should affect female-female competition for mates, and (iii) that these differences in competition

should affect female reproductive physiology (this is what the current manuscript actually reports on).

The paper is very well written and covers an interesting topic. However, I feel that it suffers from somewhat self-fulfilling, circular argumentation in the abstract, introduction, and discussion. While the language is very moderate, I still feel that some tweaking in the reasoning could improve the manuscript. As I see it, there are ecological observations in the literature that reasonably allow to speculate that there exist differences in the OSR between the two populations studied here. The current manuscript then adds another piece of supporting (circumstantial) evidence in this direction by showing that physiological investment into reproduction differs between these populations in a way consistent with the prediction of differences in the OSR. The problem is that the authors then (at times) take their findings as if they were proof of OSR differences.

For example, the authors state that they compare 'body condition and gonadal investment of fish from two populations [...]' and that they 'expect more ready-to-mate females than males' in one of them (abstract, ll. 18-21). However, the latter is never really reported on, because the authors only analyse almost even sex ratios in both populations (cf. l. 69 ff.). As the authors themselves state, 'a direct quantification of the OSR' is needed (ll. 204-205).

If the authors were to carefully check where they potentially confuse the reader over what their data can actually proof (population differences in reproductive investment and overall gross physiology) and for what their data lend additional (albeit not final) evidence (population differences in the OSR), I would be very happy.

A second main criticism I would offer is the lack of information on male reproductive investment, e.g. testes size. One could argue that a test of physiological adaptations to OSRs skewed in opposite directions would be best tested by comparing intra-population differences in reproductive investment between the sexes. In other words, rather than the inter-population intra-sexual comparison that the authors offer (for females only), one could argue for an inter-sexual intra-population comparison of reproductive traits. If the authors could address this in their methods and discussion and outline why they chose to look at females only, I would be even happier.

Other than that, I have only some minor suggestions/comments that can be found at the end of this document.

Minor comments:

l. 15: I am unsure what you mean by 'opposite-sex competition', please clarify or reword.

l. 62: please spell out 'PSU' when using it for the first time

l. 72: If you could specify how these fish were selected and what the initial sample (catch) was, that might be useful to interpret your findings.

l. 121: replace 'which' with 'who'

l. 133: 'due to its'

l. 160: replace 'reasonably' with 'arguably'

l. 173: remove highlighting

ll. 182-183: please rephrase for clarity

l. 195: replace 'to' with 'for'

l. 201: replace 'female fecundity' with 'female reproductive investment'

l. 203: replace 'be affecting' with 'in effect'

l. 314+: Figure 1. Please consider replacing your bar graph with a different form of data presentation (e.g. box plots). If you decide to stick with bar graphs, please also state what the error bars indicate. Please see Weissgerber et al. 2015 (Beyond bar and line graphs: time for a new data presentation paradigm, PLOS Biology, 13, e1002128) for good reasons why bar graphs are undesirable.

Decision letter (RSOS-181460.R0)

19-Nov-2018

Dear Ms García-Berro Navarro:

Manuscript ID RSOS-181460 entitled "Understanding resource driven female-female competition: ovary and liver size in sand gobies" which you submitted to Royal Society Open Science, has been reviewed. The comments from reviewers are included at the bottom of this letter.

In view of the criticisms of the reviewers, the manuscript has been rejected in its current form. However, a new manuscript may be submitted which takes into consideration these comments.

Please note that resubmitting your manuscript does not guarantee eventual acceptance, and that your resubmission will be subject to peer review before a decision is made.

Your resubmitted manuscript should be submitted by 19-May-2019. If you are unable to submit by this date please contact the Editorial Office.

Please note that Royal Society Open Science will introduce article processing charges for all new submissions received from 1 January 2018. Charges will also apply to papers transferred to Royal Society Open Science from other Royal Society Publishing journals, as well as papers submitted as part of our collaboration with the Royal Society of Chemistry (<http://rsos.royalsocietypublishing.org/chemistry>). If your manuscript is submitted and accepted for publication after 1 Jan 2018, you will be asked to pay the article processing charge, unless you request a waiver and this is approved by Royal Society Publishing. You can find out more about the charges at <http://rsos.royalsocietypublishing.org/page/charges>. Should you have any queries, please contact openscience@royalsociety.org.

Kind regards,
Royal Society Open Science Editorial Office
Royal Society Open Science

on behalf of Dr Kristina Sefc (Associate Editor) and Professor Kevin Padian (Subject Editor)
openscience@royalsociety.org

Subject Editor Comments:

The reviewers find interesting points in this manuscript but are disturbed by several issues. One does not feel that two populations are sufficient to establish causality, and there is the problem of controlling or eliminating other variables at either site. We will consider a resubmission but please make sure, if you resubmit, to address fully the very substantial comments in the reviews. Thanks for considering RSOS.

Associate Editor Comments to Author (Dr Kristina Sefc):

Associate Editor: 1

Comments to the Author:

Both reviewers suggest that the discussion of the manuscript should be more attentive to the limitations of the study and clearer about what the data can (or cannot) support. I trust that this will be possible in a careful revision.

Best regards, Kristina Sefc

Reviewers' Comments to Author:

Reviewer: 1

Comments to the Author(s)

This paper uses a well-established study system (sand goby fish) with a contrast in nesting sites between two regions of its range: 1. high density of nesting sites in the North Sea) and 2. their shortage in the Baltic Sea. This contrast has been used for many previous studies on sexual selection, providing strong background information to which the current study can relate. Baltic and North Seas certainly differ in many other aspects and one of them is contrasting food availability. Based on that, the authors predict (or expect as they phrase it) more female-female competition for reproductive opportunities in the well fed but nest site limited females in the Baltic. Indeed, they found heavier ovaries and liver in Baltic sand goby females, with similar, though weaker trend in the liver of males (testes were not measured).

It is a straightforward paper, with adequate sample size. The finding is not surprising (well-fed females have larger ovaries and liver – a fat storage organ in many fishes) and it provides the main strength and weakness of the study.

Strength: outcome is clear-cut; contrast between two populations provide difference that is then discussed.

Weakness: comparing only two contrasting populations. It may be read as a preliminary insight into a broader interesting study and the authors do not help to avoid this feeling. For example, in discussion it is clear that they struggle to discount alternative explanations. In many cases they succeed (the last paragraph) but most of discussion is composed in taking one potential explanation after another and discussing its potential to explain that contrast between the populations. A more useful approach would be to perform a follow-up experimental work to test competing hypotheses arising from this study.

Comments:

Introduction is succinct and well written.

Methods would certainly deserve more details on the contrast in prey abundance, density etc. This is critical for the paper but the general reference to an inventory of benthic communities should be clarified in what aspect the two sites differ (food abundance, availability, size, diversity, presence (or abundance) of one particularly important prey taxon etc.). It is addressed in part in discussion (line 122) but in a very vague way – “our own observations ... mysids extremely abundant...”

The same applies to nest site availability – is it 20% difference? Or 10 fold difference? I can imagine it is quantified in the papers that authors refer to, but reader needs to get this simple information without downloading other papers.

Results are fine, though more information is needed (perhaps in methods) on why “absolute measures” were tested in a different way and how the two measures used are useful.

Discussion rests on explaining potential alternative explanations. In some parts, it reads overly defensive. As a reader I would feel that referees made authors to write all those long paragraphs. It is of course very good to list all alternative factors that may be behind observed contrasts, but one or two sentences for discounting them (if you have strong arguments against, and in many cases you do) would suffice.

An example of this overly defensive style: Line 118, ...suggesting that female-female competition for mating opportunities may be stronger in the Baltic... - the three cited references arrived to the same conclusion, so I think it is a further confirmation rather than “suggesting that”

One particular case that made me wonder why this has not been part of the study (among others) is paragraph starting on line 145. It is on egg size in female with large/small ovaries. When ovary mass was measured, it was very easy to measure a subset of eggs per female. Why at least this has not been done?

Table 1. Why morphological traits? Perhaps life history traits? I think morphological denotes body dimensions etc.

Also, mean values are given for untransformed data – why not back-transformed data when ln- and sqrt-transformation was used in the tests?

Reviewer: 2

Comments to the Author(s)

In their manuscript ‘Understanding resource driven female-female competition: ovary and liver size in sand gobies’ (RSOS-181460) the authors report on their investigation of physiological differences between two populations of sand gobies. Because these populations differ in ecological traits related to reproduction, the authors focus on population-specific characteristics of female reproductive investment. Crucially, they state that one of the populations features traits that should make reproduction relatively more easy for males (many nest sites), while the other should make reproduction relatively more easy for females (high food abundance). They predict that these differences should (i) alter the operational sex ratio, (ii) these differences in the OSR should affect female-female competition for mates, and (iii) that these differences in competition

should affect female reproductive physiology (this is what the current manuscript actually reports on).

The paper is very well written and covers an interesting topic. However, I feel that it suffers from somewhat self-fulfilling, circular argumentation in the abstract, introduction, and discussion. While the language is very moderate, I still feel that some tweaking in the reasoning could improve the manuscript. As I see it, there are ecological observations in the literature that reasonably allow to speculate that there exist differences in the OSR between the two populations studied here. The current manuscript then adds another piece of supporting (circumstantial) evidence in this direction by showing that physiological investment into reproduction differs between these populations in a way consistent with the prediction of differences in the OSR. The problem is that the authors then (at times) take their findings as if they were proof of OSR differences.

For example, the authors state that they compare 'body condition and gonadal investment of fish from two populations [...]' and that they 'expect more ready-to-mate females than males' in one of them (abstract, ll. 18-21). However, the latter is never really reported on, because the authors only analyse almost even sex ratios in both populations (cf. l. 69 ff.). As the authors themselves state, 'a direct quantification of the OSR' is needed (ll. 204-205).

If the authors were to carefully check where they potentially confuse the reader over what their data can actually proof (population differences in reproductive investment and overall gross physiology) and for what their data lend additional (albeit not final) evidence (population differences in the OSR), I would be very happy.

A second main criticism I would offer is the lack of information on male reproductive investment, e.g. testes size. One could argue that a test of physiological adaptations to OSRs skewed in opposite directions would be best tested by comparing intra-population differences in reproductive investment between the sexes. In other words, rather than the inter-population intra-sexual comparison that the authors offer (for females only), one could argue for an inter-sexual intra-population comparison of reproductive traits. If the authors could address this in their methods and discussion and outline why they chose to look at females only, I would be even happier.

Other than that, I have only some minor suggestions/comments that can be found at the end of this document.

Minor comments:

l. 15: I am unsure what you mean by 'opposite-sex competition', please clarify or reword.

l. 62: please spell out 'PSU' when using it for the first time

l. 72: If you could specify how these fish were selected and what the initial sample (catch) was, that might be useful to interpret your findings.

l. 121: replace 'which' with 'who'

l. 133: 'due to its'

l. 160: replace 'reasonably' with 'arguably'

l. 173: remove highlighting

ll. 182-183: please rephrase for clarity

l. 195: replace 'to' with 'for'

l. 201: replace 'female fecundity' with 'female reproductive investment'

l. 203: replace 'be affecting' with 'in effect'

l. 314+: Figure 1. Please consider replacing your bar graph with a different form of data presentation (e.g. box plots). If you decide to stick with bar graphs, please also state what the error bars indicate. Please see Weissgerber et al. 2015 (Beyond bar and line graphs: time for a new data presentation paradigm, PLOS Biology, 13, e1002128) for good reasons why bar graphs are undesirable.

Author's Response to Decision Letter for (RSOS-181460.R0)

See Appendix A.

RSOS-190886.R0

Review form: Reviewer 2

Is the manuscript scientifically sound in its present form?

Yes

Are the interpretations and conclusions justified by the results?

Yes

Is the language acceptable?

Yes

Is it clear how to access all supporting data?

Yes

Do you have any ethical concerns with this paper?

No

Have you any concerns about statistical analyses in this paper?

No

Recommendation?

Accept with minor revision (please list in comments)

Comments to the Author(s)

In their manuscript 'Understanding resource driven female-female competition: ovary and liver size in sand gobies', the authors report on their results when comparing the physiology of two populations of sand gobies that differ in important ecological factors impacting reproduction (food and nests). This is a revised version of the manuscript and I had provided comments on an earlier draft. I find that most of my previous points have been addressed satisfactorily and judge the manuscript as improved and more clear especially in its reasoning, methods, and results. However, I still take issue with how the authors interpret their data: on page 27 they state that their 'result indirectly means that a larger proportion of females were ready-to-spawn' – there is

no report on proportions of 'ready-to-mate females' in the manuscript, but only evidence that females in one population had larger livers and larger ovaries, and relatively higher investment into reproductive physiology (Table 1). I think that it is misleading to interpret these findings in the way the authors do, unless they also provide evidence that a certain measure allows for the distinction between ready-to-mate and not-ready-to-mate females from which they then calculate the proportion they found in the respective populations (i.e. how many of the 18 and 31 females from the respective populations were considered ready-to-mate based on a given criterion and how many were not?). The authors indicate that they interpret their data in that way in the response letter, but as it stands, I see evidence for a greater overall reproductive investment and - potential in females from one population, but no 'indirect evidence' for differential proportions in ready-to-mate females. If the authors were to change the first paragraph of their discussion to be more in line with their last paragraph (page 30), I believe that this would be helpful. Finally, I thank the authors for changing their figure 1 for one that I find much more informative, but would request that they include a figure caption that also explains what is included in the boxes and whiskers (interquartiles, means, 95% Cis?), how outliers (points) were defined, and what the stars refer to (significance?).

Review form: Reviewer 3

Is the manuscript scientifically sound in its present form?

No

Are the interpretations and conclusions justified by the results?

No

Is the language acceptable?

Yes

Is it clear how to access all supporting data?

Yes

Do you have any ethical concerns with this paper?

No

Have you any concerns about statistical analyses in this paper?

No

Recommendation?

Major revision is needed (please make suggestions in comments)

Comments to the Author(s)

The manuscript (MS) by Aurora García-Berro and Charlotta Kvarnemo titled 'Understanding resource driven female-female competition: ovary and liver size in sand gobies' tests hypothesis that resource availability and especially food availability in females can drive OSR bias and therefore intrasexual competition for mating partners. The authors compare condition (liver mass) and female reproductive allocation (ovary size) between two geographically distant populations of small marine fish. The two study sites were previously shown to have different territory and resource availability. In the present study, the two populations of sand goby differ in female condition (liver mass) and ovary mass. The authors interpret their findings in the context of female-female competition for males and also discuss other potential factors that may have affected their findings.

The MS is well written and structured. The main research question originates from the combination of competition for sexual mates and theory of life-history evolution, and to apply it in this study system appears very interesting. More importantly, the authors test their hypothesis in the wild. Introduction builds well the idea behind the study and identifies main aims. While Methods are described appropriately and data analysis is adequate, the study design is weak to address the proposed hypothesis directly, with the first prediction not even tested. Results are presented comprehensively, but ignore significant interaction between sex and population in liver mass analysis. Discussion carefully considers various alternative explanations, but conclusion of the MS remains severely undermined by insufficient data. The interpretation of the results also ignores low number of replications collected (2). Because of these shortcomings, I suggest the authors elaborate their discussion with regard to the main issues I see in the MS: low study replication size ($N = 2$), unmeasured ecological factors (food availability, population density and territory density) and more parsimonious explanation of their main result (high condition of females in Tvärminne population). To leave some space, the last paragraph of the Discussion is only an interesting speculation with no value added and can be deleted.

The main points:

Low study replication size - The data on sexual selection from field are still scarce. Here, performing study on several replicated sites (i.e. with different populations) is a must to be able to conclude on a generality of the findings. The amount of environmental interpopulation noise (residual variation) can be accounted for only by then. This is even more pressing in studies conducted in natural systems compared to experiments in controlled/laboratory conditions. I admit this is a challenging task. After the research is done, there is not much to do. The comparison of two populations should, always be taken with caution and awareness of stochastic bias between these two. I suggest to the authors to mention the limitation of the present study sample size in the discussion and refrain from bold conclusions based on the current findings as they may be just local/temporal artefacts.

Unmeasured ecological factors - In addition to the above mentioned, the two study sites are supposed to contrast in territory availability to males and food abundance. These two focal parameters presumably affecting OSR in sand goby were not measured in the present study. The authors refer to previous work, but the studies on territory availability are more than 20 years old and I wonder whether there is no more recent data available. Regarding the food abundance, they suggest that it 'is likely to be lower' and offer their own observation as the proximate measure. Moreover, the study on the macro-invertebrate biomass in the Baltic Sea (Gogina et al. 2016) suggest that the two sites are quite similar and no clear difference is apparent between the two sites from the Supplementary Electronic Material. One of the important, yet unmeasured, parameters is population density that is nowhere to be mentioned throughout the MS. Again, I think all these need to be discussed more.

Parsimonious explanation of their main result: What makes the authors think that there is an added effect of female-female competition on ovary mass above that of high food abundance? Females from the population with more food (Tvärminne) are in better condition, so they can simply invest more into reproduction. The logic behind the current and future reproduction conflict is clear, but the data do not support it. I would expect an interaction in the relationship between body mass and ovary mass - solid line (Tvärminne) in Figure 2 to be steeper than the dashed (Bökevik); suggesting females with fuller ovaries cannot get rid of the eggs in Tvärminne population. Females in fish can, however, also resorb unspawned eggs to save part of the invested energy. This also calls for more detailed elaboration on sand goby reproductive physiology and biology, e.g. how many clutches do females lay in each population.

The resource-availability effect on OSR through ready-to-mate individuals appears only in Introduction, including the first prediction of 'more ready-to-mate females (and hence a more female-biased OSR)' [Introduction, last paragraph] in the high food population. The proportion of ready-to-mate females was, however, not compared between the two populations. The data do not suggest that 'our result indirectly means that a larger proportion of females were ready-to-

spawning the Baltic Sea' [Discussion, first paragraph]. This is a construction that requires better explanation.

Decision letter (RSOS-190886.R0)

13-Jun-2019

Dear Ms García-Berro Navarro,

The Subject Editor assigned to your paper ("Understanding resource driven female-female competition: ovary and liver size in sand gobies") has now received comments from reviewers. We would like you to revise your paper in accordance with the referee and Associate Editor suggestions which can be found below (not including confidential reports to the Editor). Please note this decision does not guarantee eventual acceptance.

Please submit a copy of your revised paper before 06-Jul-2019. Please note that the revision deadline will expire at 00.00am on this date. If we do not hear from you within this time then it will be assumed that the paper has been withdrawn. In exceptional circumstances, extensions may be possible if agreed with the Editorial Office in advance. We do not allow multiple rounds of revision so we urge you to make every effort to fully address all of the comments at this stage. If deemed necessary by the Editors, your manuscript will be sent back to one or more of the original reviewers for assessment. If the original reviewers are not available we may invite new reviewers.

When submitting your revised manuscript, you must respond to the comments made by the referees and upload a file "Response to Referees" in "Section 6 - File Upload". Please use this to document how you have responded to each of the comments, and the adjustments you have made. In order to expedite the processing of the revised manuscript, please be as specific as possible in your response.

- Ethics statement

- Data accessibility

It is a condition of publication that all supporting data are made available either as supplementary information or preferably in a suitable permanent repository. The data accessibility section should state where the article's supporting data can be accessed. This section should also include details, where possible of where to access other relevant research materials

such as statistical tools, protocols, software etc can be accessed. If the data has been deposited in an external repository this section should list the database, accession number and link to the DOI for all data from the article that has been made publicly available. Data sets that have been deposited in an external repository and have a DOI should also be appropriately cited in the manuscript and included in the reference list.

If you wish to submit your supporting data or code to Dryad (<http://datadryad.org/>), or modify your current submission to dryad, please use the following link:
<http://datadryad.org/submit?journalID=RSOS&manu=RSOS-190886>

- **Competing interests**

- **Authors' contributions**

- **Acknowledgements**

- **Funding statement**

on behalf of Dr Kristina Sefc (Associate Editor) and Kevin Padian (Subject Editor)
openscience@royalsociety.org

Associate Editor Comments to Author (Dr Kristina Sefc):

One of the reviewers has seen the manuscript before and largely agrees with the revision. One issue raised by both reviewers is lack of support for the claim about 'ready to mate' females. The second reviewer has not seen the manuscript before and is concerned about the low number of replications and unmeasured ecological factors which might produce the observed result. It is unlikely that the study design can be improved at this point. Following the first round of review, the manuscript has already been revised to address similar criticism, but it seems necessary to carry this further and address these issues more explicitly in the discussion and reflect them in the formulation of the conclusions.

Reviewer comments to Author:

Reviewer: 2

Comments to the Author(s)

In their manuscript 'Understanding resource driven female-female competition: ovary and liver size in sand gobies', the authors report on their results when comparing the physiology of two populations of sand gobies that differ in important ecological factors impacting reproduction (food and nests). This is a revised version of the manuscript and I had provided comments on an earlier draft. I find that most of my previous points have been addressed satisfactorily and judge the manuscript as improved and more clear especially in its reasoning, methods, and results. However, I still take issue with how the authors interpret their data: on page 27 they state that their 'result indirectly means that a larger proportion of females were ready-to-spawn' - there is no report on proportions of 'ready-to-mate females' in the manuscript, but only evidence that females in one population had larger livers and larger ovaries, and relatively higher investment into reproductive physiology (Table 1). I think that it is misleading to interpret these findings in the way the authors do, unless they also provide evidence that a certain measure allows for the distinction between ready-to-mate and not-ready-to-mate females from which they then calculate the proportion they found in the respective populations (i.e. how many of the 18 and 31 females from the respective populations were considered ready-to-mate based on a given criterion and how many were not?). The authors indicate that they interpret their data in that way in the response letter, but as it stands, I see evidence for a greater overall reproductive investment and -potential in females from one population, but no 'indirect evidence' for differential proportions in ready-to-mate females. If the authors were to change the first paragraph of their discussion to be more in line with their last paragraph (page 30), I believe that this would be helpful. Finally, I thank the authors for changing their figure 1 for one that I find much more informative, but would request that they include a figure caption that also explains what is included in the boxes and whiskers (interquartiles, means, 95% Cis?), how outliers (points) were defined, and what the stars refer to (significance?).

Reviewer: 3

Comments to the Author(s)

The manuscript (MS) by Aurora García-Berro and Charlotta Kvarnemo titled 'Understanding resource driven female-female competition: ovary and liver size in sand gobies' tests hypothesis that resource availability and especially food availability in females can drive OSR bias and therefore intrasexual competition for mating partners. The authors compare condition (liver mass) and female reproductive allocation (ovary size) between two geographically distant populations of small marine fish. The two study sites were previously shown to have different

territory and resource availability. In the present study, the two populations of sand goby differ in female condition (liver mass) and ovary mass. The authors interpret their findings in the context of female-female competition for males and also discuss other potential factors that may have affected their findings.

The MS is well written and structured. The main research question originates from the combination of competition for sexual mates and theory of life-history evolution, and to apply it in this study system appears very interesting. More importantly, the authors test their hypothesis in the wild. Introduction builds well the idea behind the study and identifies main aims. While Methods are described appropriately and data analysis is adequate, the study design is weak to address the proposed hypothesis directly, with the first prediction not even tested. Results are presented comprehensively, but ignore significant interaction between sex and population in liver mass analysis. Discussion carefully considers various alternative explanations, but conclusion of the MS remains severely undermined by insufficient data. The interpretation of the results also ignores low number of replications collected (2). Because of these shortcomings, I suggest the authors elaborate their discussion with regard to the main issues I see in the MS: low study replication size ($N = 2$), unmeasured ecological factors (food availability, population density and territory density) and more parsimonious explanation of their main result (high condition of females in Tvärminne population). To leave some space, the last paragraph of the Discussion is only an interesting speculation with no value added and can be deleted.

The main points:

Low study replication size - The data on sexual selection from field are still scarce. Here, performing study on several replicated sites (i.e. with different populations) is a must to be able to conclude on a generality of the findings. The amount of environmental interpopulation noise (residual variation) can be accounted for only by then. This is even more pressing in studies conducted in natural systems compared to experiments in controlled/laboratory conditions. I admit this is a challenging task. After the research is done, there is not much to do. The comparison of two populations should, always be taken with caution and awareness of stochastic bias between these two. I suggest to the authors to mention the limitation of the present study sample size in the discussion and refrain from bold conclusions based on the current findings as they may be just local/temporal artefacts.

Unmeasured ecological factors - In addition to the above mentioned, the two study sites are supposed to contrast in territory availability to males and food abundance. These two focal parameters presumably affecting OSR in sand goby were not measured in the present study. The authors refer to previous work, but the studies on territory availability are more than 20 years old and I wonder whether there is no more recent data available. Regarding the food abundance, they suggest that it 'is likely to be lower' and offer their own observation as the proximate measure. Moreover, the study on the macro-invertebrate biomass in the Baltic Sea (Gogina et al. 2016) suggest that the two sites are quite similar and no clear difference is apparent between the two sites from the Supplementary Electronic Material. One of the important, yet unmeasured, parameters is population density that is nowhere to be mentioned throughout the MS. Again, I think all these need to be discussed more.

Parsimonious explanation of their main result: What makes the authors think that there is an added effect of female-female competition on ovary mass above that of high food abundance? Females from the population with more food (Tvärminne) are in better condition, so they can simply invest more into reproduction. The logic behind the current and future reproduction conflict is clear, but the data do not support it. I would expect an interaction in the relationship between body mass and ovary mass - solid line (Tvärminne) in Figure 2 to be steeper than the dashed (Bökevik); suggesting females with fuller ovaries cannot get rid of the eggs in Tvärminne population. Females in fish can, however, also resorb unspawned eggs to save part of the invested energy. This also calls for more detailed elaboration on sand goby reproductive physiology and biology, e.g. how many clutches do females lay in each population.

The resource-availability effect on OSR through ready-to-mate individuals appears only in Introduction, including the first prediction of 'more ready-to-mate females (and hence a more female-biased OSR)' [Introduction, last paragraph] in the high food population. The proportion of ready-to-mate females was, however, not compared between the two populations. The data do not suggest that 'our result indirectly means that a larger proportion of females were ready-to-spawning the Baltic Sea' [Discussion, first paragraph]. This is a construction that requires better explanation.

Author's Response to Decision Letter for (RSOS-190886.R0)

See Appendix B.

RSOS-190886.R1 (Revision)

Review form: Reviewer 2

Is the manuscript scientifically sound in its present form?

Yes

Are the interpretations and conclusions justified by the results?

Yes

Is the language acceptable?

Yes

Do you have any ethical concerns with this paper?

No

Have you any concerns about statistical analyses in this paper?

No

Recommendation?

Accept as is

Comments to the Author(s)

In their manuscripts "Understanding resource driven female-female competition: ovary and liver size in sand gobies" (RSOS-190886), the authors report on a field study they conducted to compare two populations of their focal species that differ in various ecological parameters linked to reproduction. The authors find that this influences physiology of both sexes, but particularly that of females. They argue that this is suggestive of differences in the level of investment into reproduction between the females, with one population likely experiencing higher levels of female-female competition than the other and favouring current over future reproduction.

This is the third time that I have reviewed this manuscript. As in the previous two rounds, the authors have appropriately responded to my previous queries. Especially the inclusion of the 'roundness' data is welcome, now justifying the authors' use of 'ready-to-mate females' as a

measure of potential sexual conflict. As such, the paper is well written, all of its claims are now supported by data, and I have no more major comments about the manuscript.

At some stage, you may want to re-visit the following section to improve clarity: P. 36, Para 2: 'Gonadal [...] hepatocytes'.

Review form: Reviewer 3

Is the manuscript scientifically sound in its present form?

Yes

Are the interpretations and conclusions justified by the results?

Yes

Is the language acceptable?

Yes

Do you have any ethical concerns with this paper?

No

Have you any concerns about statistical analyses in this paper?

No

Recommendation?

Accept as is

Comments to the Author(s)

I think the Authors have put substantial effort into strengthening their MS and I can recommend the current improved version for publication.

Decision letter (RSOS-190886.R1)

31-Jul-2019

Dear Ms García-Berro Navarro,

I am pleased to inform you that your manuscript entitled "Understanding resource driven female-female competition: ovary and liver size in sand gobies" is now accepted for publication in Royal Society Open Science.

Royal Society Open Science operates under a continuous publication model (<http://bit.ly/cpFAQ>). Your article will be published straight into the next open issue and this will be the final version of the paper. As such, it can be cited immediately by other researchers.

As the issue version of your paper will be the only version to be published I would advise you to check your proofs thoroughly as changes cannot be made once the paper is published.

on behalf of Dr Kristina Sefc (Associate Editor) and Kevin Padian (Subject Editor)
openscience@royalsociety.org

Associate Editor Comments to Author (Dr Kristina Sefc):

I'd like to thank the authors for their efforts put into revising the manuscript. Both reviewers are happy with the current version (reviewer 2 offers one more suggestion for wording). Best regards, Kristina Sefc

Reviewer comments to Author:
Reviewer: 3

Comments to the Author(s)
I think the Authors have put substantial effort into strengthening their MS and I can recommend the current improved version for publication.

Reviewer: 2

Comments to the Author(s)
In their manuscripts "Understanding resource driven female-female competition: ovary and liver size in sand gobies" (RSOS-190886), the authors report on a field study they conducted to compare two populations of their focal species that differ in various ecological parameters linked to reproduction. The authors find that this influences physiology of both sexes, but particularly that of females. They argue that this is suggestive of differences in the level of investment into reproduction between the females, with one population likely experiencing higher levels of female-female competition than the other and favouring current over future reproduction.

This is the third time that I have reviewed this manuscript. As in the previous two rounds, the authors have appropriately responded to my previous queries. Especially the inclusion of the 'roundness' data is welcome, now justifying the authors' use of 'ready-to-mate females' as a measure of potential sexual conflict. As such, the paper is well written, all of its claims are now supported by data, and I have no more major comments about the manuscript.

At some stage, you may want to re-visit the following section to improve clarity: P. 36, Para 2: 'Gonadal [...] hepatocytes'.

Appendix A

Dear Associate Editor Dr Kristina Sefc and Subject Editor Professor Kevin Padian,

Thank you for letting us resubmit our manuscript “Understanding resource driven female-female competition: ovary and liver size in sand gobies” to Royal Society Open Science. We appreciate all the comments - very useful and well considered. Below we list how we have dealt with each of them. We hope you will now find the manuscript ready to be accepted for publication.

Yours sincerely,

Aurora García-Berro and Charlotta Kvarnemo

Subject Editor Comments:

The reviewers find interesting points in this manuscript but are disturbed by several issues. One does not feel that two populations are sufficient to establish causality, and there is the problem of controlling or eliminating other variables at either site. We will consider a resubmission but please make sure, if you resubmit, to address fully the very substantial comments in the reviews. Thanks for considering RSOS.

REPLY: We are well aware of the limitations of our study, and we had (in response to previous referee comments that we got from Biology Letters) already tried to highlight that. However, we have now carefully revised the manuscript again, and hopefully found the right balance between confidence and caution.

Associate Editor Comments to Author (Dr Kristina Sefc):

Associate Editor: 1

Comments to the Author:

Both reviewers suggest that the discussion of the manuscript should be more attentive to the limitations of the study and clearer about what the data can (or cannot) support. I trust that this will be possible in a careful revision.

Best regards, Kristina Sefc

REPLY: Please see our response above.

Reviewers' Comments to Author:

Reviewer: 1

Comments to the Author(s)

This paper uses a well-established study system (sand goby fish) with a contrast in nesting sites between two regions of its range: 1. high density of nesting sites in the North Sea) and 2. their shortage in the Baltic Sea. This contrast has been used for many previous studies on sexual selection, providing strong background information to which the current study can relate. Baltic and North Seas certainly differ in many other aspects and one of them is contrasting food availability. Based on that, the authors predict (or expect as they phrase it) more female-female competition for reproductive opportunities in the well fed but nest site limited females in the Baltic. Indeed, they found heavier ovaries and liver in Baltic sand goby females, with similar, though weaker trend in the liver of males (testes were not measured).

REPLY: We have changed “expect” to “predict” in most cases.

It is a straightforward paper, with adequate sample size. The finding is not surprising (well-fed females have larger ovaries and liver – a fat storage organ in many fishes) and it provides the main strength and weakness of the study.

Strength: outcome is clear-cut; contrast between two populations provides difference that is then discussed.

Weakness: comparing only two contrasting populations. It may be read as a preliminary insight into a broader interesting study and the authors do not help to avoid this feeling. For example, in discussion it is clear that they struggle to discount alternative explanations. In many cases they succeed (the last paragraph) but most of discussion is composed in taking one potential explanation after another and discussing its potential to explain that contrast between the populations. A more useful approach would be to perform a follow-up experimental work to test competing hypotheses arising from this study.

REPLY: This manuscript was transferred to RSOS from Biology Letters, and the long list of alternative explanations was added in response to referee comments given by Biology Letters. We agree that a better way to handle them would be to do a follow-up study, testing these alternative explanations. In response to this comment, the list is shortened, and (although we find all the explanations unlikely to explain the found difference in ovary size) we now state that they need to be tested in future studies.

Regarding the shortened list of alternative explanations: We deleted point 1 (effect of body size on ovary size), since (a) there wasn't any difference in body size between the two populations in our sample, and (b) we used log-transformed values and body mass as covariate, so it wouldn't had been an issue even if there were a difference. The other points are now explained using slightly fewer words.

Comments:

Introduction is succinct and well written.

Methods would certainly deserve more details on the contrast in prey abundance, density etc. This is critical for the paper but the general reference to an inventory of benthic communities should be clarified in what aspect the two sites differ (food abundance, availability, size, diversity, presence (or abundance) of one particularly important prey taxon etc.). It is addressed in part in discussion (line 122) but in a very vague way – “our own observations ... mysids extremely abundant...”

REPLY: Good point. We have now added more detail to food availability in the method section, and added a table with values of abundance or biomass of benthic macrofauna representative of the two study areas.

The same applies to nest site availability – is it 20% difference? Or 10 fold difference? I can imagine it is quantified in the papers that authors refer to, but reader needs to get this simple information without downloading other papers.

REPLY: The nest site availability found in Forsgren et al. (1996) was 48 times higher at the Swedish west coast site than at the Baltic site (mean \pm SE: 9.6 ± 2.3 nests per m² in Bökevik Bay, and 0.2 ± 0.03 nests per m² at Tvärminne). This information is now added to the manuscript.

Results are fine, though more information is needed (perhaps in methods) on why “absolute measures” were tested in a different way and how the two measures used are useful.

REPLY: We clearly prefer using absolute values of ovaries and liver, with somatic body mass included as a covariate in ANCOVA, to ratios such as GSI and HSI. However, since GSI and HSI are extremely common in other studies, particular in the fish literature, it is justified to provide those measures as well, “to allow comparison with other studies” as we put it. We have now added more detail to the methods to justify this.

Discussion rests on explaining potential alternative explanations. In some parts, it reads overly defensive. As a reader I would feel that referees made authors to write all those long paragraphs. It is of course very good to list all alternative factors that may be behind observed contrasts, but one or two sentences for discounting them (if you have strong arguments against, and in many cases you do) would suffice.

REPLY: Yes, that is correct. As mentioned above, they were written in response to previous referee comments. However, we have now happily reduced them to a shorter text. In particular, we deleted point 1 (effect of body size on ovary size), since (a) there wasn't any difference in body size between the two populations in our sample, and (b) we used log-transformed values and body mass as covariate, so it wouldn't had been an issue even if there were a difference. We have also shortened the other points.

An example of this overly defensive style: Line 118, ...suggesting that female-female competition for mating opportunities may be stronger in the Baltic... - the three cited references arrived to the same conclusion, so I think it is a further confirmation rather than “suggesting that”

REPLY: Spot on. Now revised.

One particular case that made me wonder why this has not been part of the study (among others) is paragraph starting on line 145. It is on egg size in female with large/small ovaries. When ovary mass was measured, it was very easy to measure a subset of eggs per female. Why at least this has not been done?

REPLY: Unfortunately, it was not done at the time of dissections. However, egg size also depends on egg maturation and since sand gobies are batch spawners, they have most of their eggs at one particular stage of maturation. Therefore, to make a fair comparison between the two regions, one would need to bring in mature females directly from the field, let them spawn in the lab, and then count eggs and measure egg size. To collect such data in a well-designed manner (ideally also measuring interspawning interval between successive clutches) is a major undertaking, especially as it will require data to be collected at two distantly located field sites and, again ideally, including fish collected from more than one population within each region. It is therefore a project for a follow-up study done in the future. We have now clarified this in the discussion. That said, even if it would be interesting to know the details of how female allocation to egg production may differ between the two sites, it is not essential for this particular study, which aim was to document if ovary size differs in the direction we had predicted based on our knowledge of nest and food availability.

Table 1. Why morphological traits? Perhaps life history traits? I think morphological denotes body dimensions etc.

REPLY: We agree that “morphological traits” might not be the best choice of words. However, we don’t think “life-history traits” is an accurate description of our measured traits either. We have therefore changed the legend to say exactly that: letting “measured traits” replace “morphological traits”.

Also, mean values are given for untransformed data – why not back-transformed data when ln- and sqrt-transformation was used in the tests?

REPLY: We have now replaced the untransformed values with back-transformed values.

Reviewer: 2
Comments to the Author(s)

In their manuscript ‘Understanding resource driven female-female competition: ovary and liver size in sand gobies’ (RSOS-181460) the authors report on their investigation of physiological differences between two populations of sand gobies. Because these populations differ in ecological traits related to reproduction, the authors focus on population-specific characteristics of female reproductive investment. Crucially, they state that one of the populations features traits that should make reproduction relatively more easy for males (many nest sites), while the other should make reproduction relatively more easy for females (high food abundance). They predict that these differences should (i) alter the operational sex ratio, (ii) these differences in the OSR should affect female-female competition for mates, and (iii) that these differences in competition should affect female reproductive physiology (this is what the current manuscript actually reports on).

The paper is very well written and covers an interesting topic.

REPLY: Thank you.

However, I feel that it suffers from somewhat self-fulfilling, circular argumentation in the abstract, introduction, and discussion. While the language is very moderate, I still feel that some tweaking in the reasoning could improve the manuscript. As I see it, there are ecological observations in the literature that reasonably allow to speculate that there exist differences in the OSR between the two populations studied here.

REPLY: OSR has never been directly quantified at the two study areas, but sure, Forsgren et al. (1996) provides data on nest site availability that allow us to speculate in that direction. We have now rewritten the first paragraph of the method section to make that clearer (together with other clarifications, that relate to comments by Reviewer 1).

The current manuscript then adds another piece of supporting (circumstantial) evidence in this direction by showing that physiological investment into reproduction differs between these populations in a way consistent with the prediction of differences in the OSR. The problem is that the authors then (at times) take their findings as if they were proof of OSR differences.

REPLY: That is correct: OSR has not been quantified, not before and not in this study, and our current data only provide circumstantial evidence in the predicted direction.

Still, since female sand gobies are batch spawners they go in and out of OSR, being in ‘time-in’ only when they have mature eggs to lay. Consequently, OSR is strongly affected by the proportion of females that are ready to spawn. Our current data show larger ovaries in the Baltic Sea, meaning there are more females in ‘time-in’ there. We have now rephrased ourselves, trying to make it clearer what we can and what we cannot say about OSR in sand gobies.

For example, the authors state that they compare ‘body condition and gonadal investment of fish from two populations [...]’ and that they ‘expect more ready-to-mate

females than males' in one of them (abstract, ll. 18-21). However, the latter is never really reported on, because the authors only analyse almost even sex ratios in both populations (cf. l. 69 ff.). As the authors themselves state, 'a direct quantification of the OSR' is needed (ll. 204-205).

REPLY: Thank you for pointing this out: It was incorrect of us to mention males in the abstract, since we have only collected useful data on females in relation to OSR, in this study. The cited sentence is now rephrased as "Comparing body condition and gonadal investment of fish from two populations (Baltic Sea: few nest-sites, much food; North Sea: many nest-sites, less food), we predict more ready-to-mate females (and hence a more female-biased OSR) in the Baltic Sea than in the North Sea."

If the authors were to carefully check where they potentially confuse the reader over what their data can actually prove (population differences in reproductive investment and overall gross physiology) and for what their data lend additional (albeit not final) evidence (population differences in the OSR), I would be very happy.

REPLY: We have carefully revised the manuscript with this good advice in mind.

A second main criticism I would offer is the lack of information on male reproductive investment, e.g. testes size. One could argue that a test of physiological adaptations to OSRs skewed in opposite directions would be best tested by comparing intra-population differences in reproductive investment between the sexes. In other words, rather than the inter-population intra-sexual comparison that the authors offer (for females only), one could argue for an inter-sexual intra-population comparison of reproductive traits. If the authors could address this in their methods and discussion and outline why they chose to look at females only, I would be even happier.

REPLY: Testes size wouldn't tell us whether a male is part of OSR or not. What determines this is instead whether the male has a nest site or not. Since the male aspect of OSR has been investigated before (Forsgren et al. 1996, Singer et al. 2006) we chose to focus on females rather than males here.

Other than that, I have only some minor suggestions/comments that can be found at the end of this document.

Kind regards

Arne Jungwirth (I always sign my reviews where I know authors' identities)

Department of Zoology

University of Cambridge

Minor comments:

l. 15: I am unsure what you mean by 'opposite-sex competition', please clarify or reword.

REPLY: Rewritten ('competition within the opposite sex for resource-holding mates')

l. 62: please spell out 'PSU' when using it for the first time

REPLY: PSU means practical salinity units. However, to make it more accessible, it is now replaced with '‰ salinity', or just '‰'.

l. 72: If you could specify how these fish were selected and what the initial sample (catch) was, that might be useful to interpret your findings.

REPLY: All the fish that were caught were used. This is now clarified ('No selective criteria were applied, i.e., all fish caught were kept.').

l. 121: replace 'which' with 'who'

REPLY: Done.

l. 133: 'due to its'

REPLY: Done.

l. 160: replace 'reasonably' with 'arguably'

REPLY: Done.

l. 173: remove highlighting

REPLY: Not found. Consider it done.

ll. 182-183: please rephrase for clarity

REPLY: Now deleted.

l. 195: replace 'to' with 'for'

REPLY: Done.

l. 201: replace 'female fecundity' with 'female reproductive investment'

REPLY: Done.

l. 203: replace 'be affecting' with 'in effect'

REPLY: Done.

l. 314+: Figure 1. Please consider replacing your bar graph with a different form of data presentation (e.g. box plots). If you decide to stick with bar graphs, please also state what the error bars indicate. Please see Weissgerber et al. 2015 (Beyond bar and line graphs: time for a new data presentation paradigm, PLOS Biology, 13, e1002128) for good reasons why bar graphs are undesirable.

REPLY: We have now replaced it with a boxplot.

References

Forsgren E, Kvarnemo C, Lindström K. 1996 Mode of sexual selection determined by resource abundance in two sand goby populations. *Evolution* **50**, 646–654.

Singer A, Kvarnemo C, Lindström K, Svensson O 2006. Genetic mating patterns studied in pools with manipulated nest site availability and two populations of *Pomatoschistus minutus*. *J Evol Ecol* **19**: 1641-1650.

Appendix B

Response to the editor

Dear Dr Kristina Sefc:

Thank you for letting us resubmit our manuscript 'Understanding resource driven female-female competition: ovary and liver size in sand gobies'. We have addressed the reviewers suggestions thoroughly both by editing the previous version of the manuscript and by adding new data to the study. We hope the manuscript is now ready for publication. New people has been involved in the confection of this version (contributions specified in the according section): Johanna Yliportimo and Kai Lindström. We have therefore agreed to share the authorship with them. We are, however, facing problems with the submission application to add Kai Lindström, as he was previously suggested as a recommended reviewer in the first submission. We also hope that this issue will be solved.

Thank you very much,
Sincerely,

Aurora García-Berro

Response to the reviewers

Comments to the Author(s)

In their manuscript 'Understanding resource driven female-female competition: ovary and liver size in sand gobies', the authors report on their results when comparing the physiology of two populations of sand gobies that differ in important ecological factors impacting reproduction (food and nests). This is a revised version of the manuscript and I had provided comments on an earlier draft. I find that most of my previous points have been addressed satisfactorily and judge the manuscript as improved and more clear especially in its reasoning, methods, and results.

Thank you.

However, I still take issue with how the authors interpret their data: on page 27 they state that their 'result indirectly means that a larger proportion of females were ready-to-spawn' – there is no report on proportions of 'ready-to-mate females' in the manuscript, but only evidence that females in one population had larger livers and larger ovaries, and relatively higher investment into reproductive physiology (Table 1). I think that it is misleading to interpret these findings in the way the authors do, unless they also provide evidence that a certain measure allows for the distinction between ready-to-mate and not-ready-to-mate females from which they then calculate the proportion they found in the respective populations (i.e. how many of the 18 and 31 females from the respective populations were considered ready-to-mate based on a given criterion and how many were not?). The authors indicate that they interpret their data in that way in the response letter, but as it stands, I see evidence for a greater overall reproductive investment and -potential in females from one population, but no 'indirect evidence' for differential proportions in ready-to-mate females. If the authors were to change the first paragraph of their discussion to be more in line with their last paragraph (page 30), I believe that this would be helpful.

Female sand gobies are batch spawners. This means they mature a clutch, gradually get rounder (=heavier ovaries), spawn, get slim, mature a new clutch... and so on. This means their roundness is a very good indicator of if they are ready-to-spawn or not. This information is now added to the introduction. In previous work, we have used a visually based score of female roundness, to predict if females are ready to spawn or not. The scale goes between 0 and 3, with values equal to or greater than 2.25 representing females that are judged to have mature eggs ready to spawn. Since this score was collected for this data set too, allowing us to provide proportion values for each of the populations of ready-to-spawn females, we have now added this information to the manuscript. We have also modified the discussion and our conclusions to embrace these changes.

Finally, I thank the authors for changing their figure 1 for one that I find much more informative, but would request that they include a figure caption that also explains what is included in the boxes and whiskers (interquartiles, means, 95% Cis?), how outliers (points) were defined, and what the stars refer to (significance?).

Thank you for suggesting us to do so. The information about whiskers, stars etc was meant to be uploaded as the figure foot, but we had to try several times and apparently did not succeed (maybe due to MAC-windows incompatibilities). We hope it is fixed now.

Comments to the Author(s)

The manuscript (MS) by Aurora García-Berro and Charlotta Kvarnemo titled 'Understanding resource driven female-female competition: ovary and liver size in sand gobies' tests hypothesis that resource availability and especially food availability in females can drive OSR bias and therefore intrasexual competition for mating partners. The authors compare condition (liver mass) and female reproductive allocation (ovary size) between two geographically distant populations of small marine fish. The two study sites were previously shown to have different territory and resource availability. In the present study, the two populations of sand goby differ in female condition (liver mass) and ovary mass. The authors interpret their findings in the context of female-female competition for males and also discuss other potential factors that may have affected their findings.

The MS is well written and structured. The main research question originates from the combination of competition for sexual mates and theory of life-history evolution, and to apply it in this study system appears very interesting. More importantly, the authors test their hypothesis in the wild. Introduction builds well the idea behind the study and identifies main aims. While Methods are described appropriately and data analysis is adequate, the study design is weak to address the proposed hypothesis directly, with the first prediction not even tested. Results are presented comprehensively, but ignore significant interaction between sex and population in liver

mass analysis. Discussion carefully considers various alternative explanations, but conclusion of the MS remains severely undermined by insufficient data. The interpretation of the results also ignores low number of replications collected (2). Because of these shortcomings, I suggest the authors elaborate their discussion with regard to the main issues I see in the MS: low study replication size (N = 2), unmeasured ecological factors (food availability, population density and territory density) and more parsimonious explanation of their main result (high condition of females in Tvärminne population). To leave some space, the last paragraph of the Discussion is only an interesting speculation with no value added and can be deleted.

Thank you for your constructive criticism. Specifically:

- **First prediction not even tested:** Good point. It is now.
- **Ignore significant interaction:** We do emphasise this interaction in the abstract, by saying that “liver (storage of energy reserves and vitellogenic compounds)” was larger in the Baltic sea “for both sexes, but particularly so for females”. However, we now return to this result and refer to it in the discussion as well.
- **Conclusion undermined by insufficient data:** We have now added more data and also tried to be even more careful with our conclusions.
- **Low study replication size:** See below.
- **Unmeasured ecological factors:** We agree this is another short-coming of the study. To improve it as much as we can, we have added new data on nest site colonisation rate and more references are added to Table S1.
- **Parsimonious explanation:** see below.

The main points:

Low study replication size - The data on sexual selection from field are still scarce. Here, performing study on several replicated sites (i.e. with different populations) is a must to be able to conclude on a generality of the findings. The amount of environmental interpopulation noise (residual variation) can be accounted for only by then. This is even more pressing in studies conducted in natural systems compared to experiments in controlled/laboratory conditions. I admit this is a challenging task. After the research is done, there is not much to do. The comparison of two populations should, always be taken with caution and awareness of stochastic bias between these two. I suggest to the authors to mention the limitation of the present study sample size in the discussion and refrain from bold conclusions based on the current findings as they may be just local/temporal artefacts.

We agree this is a short-coming of the study, and we now point it out as such in the discussion. Several sites in each area would be much more informative, but that would signify a major undertaking. We have now, as suggested, tried to rephrase the discussion at different points in regard to our sample size limitation.

Unmeasured ecological factors - In addition to the above mentioned, the two study sites are supposed to contrast in territory availability to males and food abundance. These two focal

parameters presumably affecting OSR in sand goby were not measured in the present study. The authors refer to previous work, but the studies on territory availability are more than 20 years old and I wonder whether there is no more recent data available.

We agree that the very central source on nest site availability and sexual selection is old (published in 1996, with its data collected in 1993), and therefore we have now added more recent data on nest site colonisation rate (collected in 2014 at Kristineberg and complemented with new data from Tvärminne 2019, collected in the same way). Importantly, these new results are strikingly similar to what was found in 1993, which shows that the information in Forsgren *et al.* (1996) is still valid. In addition, we cite a Mück & Heubel (2018), which further supports a more general conclusion that nest site availability differs markedly between the two areas.

Regarding the food abundance, they suggest that it is likely to be lower' and offer their own observation as the proximate measure. Moreover, the study on the macro-invertebrate biomass in the Baltic Sea (Gogina *et al.* 2016) suggest that the two sites are quite similar and no clear difference is apparent between the two sites from the Supplementary Electronic Material.

We were not able to directly measure food availability in this study; instead we performed an exhaustive literature search for the localities and all information is contained in Table S1. Although the differences reported in Table S1 may be moderate, this information still supports our own impression that food availability is higher at the Baltic Sea site. We also base our argument on the fact that Baltic Sea is becoming eutrophic (Andersen *et al.* 2017), and thus is able to sustain larger populations of polychaetes and small crustaceans in comparison to the North Sea area. We address this further in the manuscript and we have extended Table S1 with new references.

One of the important, yet unmeasured, parameters is population density that is nowhere to be mentioned throughout the MS. Again, I think all these need to be discussed more.

Unfortunately we don't have a direct measure of fish density. In the newly added nest site colonisation experiment, however, relative density effects can be reflected in the nest occupancy rates. At the Baltic site (Tvärminne), nest were colonised much faster and to a higher extent than at the North Sea site (Bökevik), suggesting that it continues to be a scarce resource and a limiting resource for male-readiness to mate at Tvärminne.

Parsimonious explanation of their main result: What makes the authors think that there is an added effect of female-female competition on ovary mass above that of high food abundance? Females from the population with more food (Tvärminne) are in better condition, so they can simply invest more into reproduction. The logic behind the current and future reproduction conflict is clear, but the data do not support it. I would expect an interaction in the relationship between body mass and ovary mass - solid line (Tvärminne) in Figure 2 to be steeper than the dashed (Bökevik); suggesting females with fuller ovaries cannot get rid of the eggs in Tvärminne population. Females in fish can, however, also resorb unspawned eggs to save part of the invested energy. This also calls for more detailed elaboration on sand goby reproductive physiology and biology, e.g. how many clutches do females lay in each population.

Yes, high food abundance is likely to be one of the main reasons behind heavier gonads in Tvärminne fish. Its main effect is to accelerate egg maturation rate, but it also affects female ability to maintain a large clutch size over successive clutches (Kvarnemo 1997). The result of this would be a higher number of females available to mate at any given time point (more ready-to-mate females: our newly added data on female roundness ('reproductive maturity index') supports this notion with significantly rounder females, and with a significantly higher proportion of females being scored as ready to spawn at Tvärminne). However, when this effect is combined with the above discussed nest site limitation (which we think is more pronounced than the difference in food availability), we expect females to face a difficulty spawning their mature eggs. Hence, we have good reasons to argue that both factors are likely to contribute to the larger ovaries found at Tvärminne, and we do not agree that food availability on its own would provide a more parsimonious explanation.

We have added some more information on sand goby reproductive biology, including interspawning intervals for the two sexes and a few words on female batch spawning and how it relates to whether a female is ready to spawn or not (=part of OSR or not). Both sexes breed repeatedly in multiple brood cycles over their single breeding season. A guess is 1-4 cycles over the season, however, we have no hard data from the field telling us how many cycles a fish goes through in the wild (in a lab study, in which females were followed for the better part of their reproductive life, most females produced 1-3 clutches, but two of the females managed to lay 7 clutches: C. Kvarnemo unpublished data). Given the uncertainty in these estimates, we think "multiple brood cycles" is enough, but if you consider it essential that we provide a number, then it should be 1-4.

As for the ability to resorb eggs, yes it is true also for sand gobies. However, doing so results in delayed reproduction, and for a species with only one (fairly short) breeding season, such a delay comes with a great cost to lifetime reproductive success.

Nevertheless, looking at the minimum ovary sizes, which we expect to belong to empty ovaries just caught after spawning, we see a difference between populations. So despite not having observed female competition, we think that a difference in ovary size is unlikely driven by food availability solely - although possible - and that something else may have helped drive fish in Tvärminne to have such heavier gonads relative to their size. We have now rephrased this text slightly to include the possibility of gonad size evolving due to high food availability as well .

The resource-availability effect on OSR through ready-to-mate individuals appears only in Introduction, including the first prediction of 'more ready-to-mate females (and hence a more female-biased OSR)' [Introduction, last paragraph] in the high food population. The proportion of ready-to-mate females was, however, not compared between the two populations. The data do not suggest that 'our result indirectly means that a larger proportion of females were ready-to-spawning the Baltic Sea' [Discussion, first paragraph]. This is a construction that requires better explanation.

This topic was also raised in the other reviewer's comments. In response to these comments, we decided to add data on female roundness ('reproductive maturity index') that was collected as part of this study, but not used previously. The benefit of including it is that it allows us to score whether a female is ready to spawn or not. This data supports our prediction in the sense that we found significantly rounder females at Tvärminne, and a significantly higher proportion of females being scored as ready to spawn at Tvärminne. We have updated our discussion to better reflect this result.